# Synthesis and synergistic antibacterial activity of rafoxanide derivative as a novel drug for potentiating colistin activity

Haotian Shao,[1] Qiange Liu,[1] Peiyi Liu,[1] Xiaoyuan Ma,[1] Zibo Li,[2] Yajun Zhai,[1] Hua Wu,[1] Dandan He,[1] Guoyu Yang,[3] Gongzheng Hu[1]

**ABSTRACT** Antimicrobial resistance poses a significant global public health challenge in the 21st century, with multidrug resistance in gram-negative bacteria being particularly severe. Colistin (COL) is considered the "last-resort" for treating such infections. Still, its efficacy is significantly reduced due to the emergence and prevalence of mobile colistin resistance (MCR)-mediated acquired drug resistance. Based on the synergistic antibacterial mechanism between the anthelmintic rafoxanide (RAF) and COL, we synthesized a novel RAF derivative (named as dikalisuan) through a carboxymethylation method, and its structure was confirmed using $^1$H NMR, $^{13}$C NMR, and HRMS. *In vitro* evaluation indicated that the derivatized product had no antibacterial activity (minimum inhibitory concentration [MIC] > 512 µg/mL). However, when used in combination with COL, it showed a significantly synergistic effect (fractional inhibitory concentration index [FICI] < 0.5) on *mcr-1*-positive and -negative resistant strains, including *Escherichia coli*, *Salmonella*, *Klebsiella pneumoniae*, and *Acinetobacter baumannii*. Notably, the synergistic bactericidal effect was optimal at a moderate concentration of 8 µg/mL. Furthermore, within the concentration range of 2 to 128 µg/mL, dikalisuan alone or in combination with COL (2 µg/mL) showed no significant hemolysis of red blood cells or cytotoxicity. This derivative provides a new strategy for developing adjuvants against multidrug-resistant bacteria and lays the foundation for the structural modification of anthelmintic salicylanilides.

**IMPORTANCE** Antimicrobial resistance in gram-negative bacteria is a critical global health threat, with colistin (COL) being a "last-resort" treatment. However, rising resistance has blunted its effectiveness. This study developed a new derivative of rafoxanide (RAF). While this new compound itself is not antibacterial, it effectively potentiates COL's power against resistant bacteria when used in combination. This strategy offers a promising approach to extend the lifespan of our current antibiotics and provides a new weapon in the fight against difficult-to-treat drug-resistant bacteria.

**KEYWORDS** Colistin, rafoxanide, carboxymethylation, multidrug-resistant bacteria

The increasing issue of antimicrobial resistance endangers the conventional regimens for treating bacterial infectious diseases (1). It is projected that approximately 1.91 million deaths attributable to antimicrobial resistance and 8.22 million deaths associated with antimicrobial resistance could occur globally in 2050 (2). The widespread emergence of multidrug-resistant bacteria has threatened the health of humans and animals, and clinical anti-infection treatment has thus fallen into a predicament. Infections caused by gram-negative bacteria can be considered difficult to treat as they are essentially resistant to many antibiotics due to their dual-membrane envelope, which prevents many antibiotics from accessing targets, thus typically limiting treatment options (3, 4). In the past 50 years, almost no new compounds have been found to be active against gram-negative bacteria (5). The lack of available options for treating multidrug-resistant

**Peer Reviewers** Yujie Hu, Key Laboratory of Food Safety Risk Assessment, Beijing, China; Innocent Afeke, University of Health and Allied Sciences, Ho, Volta Region, Ghana

Address correspondence to Gongzheng Hu, yaolilab@126.com, or Guoyu Yang, yangguoyulxy@henau.edu.cn.

Haotian Shao, Qiange Liu, and Peiyi Liu contributed equally to this article. Author order was determined by the sequence of research participation and contribution.

The authors declare no conflict of interest.

See the funding table on p. 9.

gram-negative bacterial infections has driven the development of new strategies to overcome existing resistance mechanisms. One proven economic and effective strategy is to combine existing antibiotics with adjuvants for treatment (6).

Colistin (COL) belongs to the polypeptide antibiotics and is commonly used to treat multidrug-resistant gram-negative bacterial infections. It is known as the "last-resort" (7, 8). It was initially isolated from *Paenibacillus polymyxa Colistinus* and was used as an intravenous antibiotic. COL targets the lipopolysaccharide (LPS) component of the cell membrane of gram-negative bacteria, disrupting membrane integrity and leading to bacterial cell death (9). Previously, it was thought that resistance to COL was intrinsic, until the first mobilized colistin resistance (MCR) gene *mcr-1* was reported in 2015 (10). The *mcr-1* gene is believed to have predated its identification, mainly attributed to COL's extensive use in animal husbandry and veterinary medicine. Due to its ability to transfer between different bacterial genera, it is often isolated from clinical samples and poses a significant threat to public health (11). After the report of *mcr-1*, an additional nine *mcr* genes have been found and given new numerical designations, namely, *mcr-2* to *mcr-10* (12). In addition to the plasmid-mediated resistance, several chromosomal resistance mechanisms were also identified, such as mutations in the *phoPQ*, *pmrAB,* and *mgrB* genes (13). Currently, *mcr-1* is the most widely spread member of the *mcr* family (14). In summary, both plasmid-mediated resistance and chromosome-mediated resistance ultimately reduce the binding ability of COL by modifying the LPS structure of the bacterial outer membrane (15).

However, due to the widespread application of COL leading to the emergence of drug resistance genes, its therapeutic effect is significantly reduced (10). Previous studies have shown that the synergistic combinations of anthelmintic salicylanilides such as rafoxanide (RAF), oxyclozanide (OXY), and closantel (CLO) with COL eradicate multidrug-resistant COL-resistant gram-negative bacteria (16). RAF is commonly used to treat trematode infections in ruminants. The research conducted by Han et al. indicates that the combination of COL and RAF destroys the inner-membrane integrity and quenches ATP synthesis, leading to inhibition of efflux pump activity. At the same time, there is also an excessive production of reactive oxygen species (ROS) (17). These mechanisms enable RAF without antibacterial activity to restore the sensitivity of antibiotic-resistant bacteria to COL.

In this study, dikalisuan was synthesized via a carboxymethylation method using RAF as the starting material, and its structure was characterized by NMR and HRMS. The antibacterial activity of the RAF derivative was explored against *Escherichia coli*, *Salmonella*, *Klebsiella pneumoniae*, and *Acinetobacter baumannii*, and the safety assessment was conducted. This study provides a reference for the chemical modification of salicylanilide drugs and offers more alternative drugs for the clinical treatment of COL-resistant gram-negative bacterial infections.

## MATERIALS AND METHODS

### Materials and reagents

A total of 15 nonduplicate COL-resistant gram-negative bacteria were selected randomly from the Pharmacology Laboratory of Henan Agricultural University, including *E. coli* (*n* = 4), *Salmonella* (*n* = 4), *K. pneumoniae* (*n* = 6), and *A. baumannii* (*n* = 1). Our laboratory previously amplified the *mcr-1* gene through polymerase chain reaction (PCR) to define whether the *mcr-1* gene was detected as positive or negative. Among them, there were seven *mcr-1*-positive strains (two strains of *E. coli*, two strains of *Salmonella*, and three strains of *K. pneumoniae*). The remaining eight strains were *mcr-1*-negative strains. Rafoxanide (Lot No. 20230916) with a purity of 99.4% was provided by Wuhan Huajiu Pharmaceutical Technology Co., Ltd. (Wuhan, China). Tetrahydrofuran, ethyl bromoacetate, and dimethyl sulfoxide-$d_6$ (DMSO-$d_6$) were purchased from Shanghai Macklin Biochemical Co., Ltd. (Shanghai, China). Hexane and anhydrous ethanol were purchased

from Sinopharm Chemical Reagent Co., Ltd. (Shanghai, China). All other chemicals and reagents were of analytical purity.

## Synthesis of dikalisuan

Dikalisuan was synthesized through a carboxymethylation method.

RAF (1 mmol, $C_{19}H_{11}Cl_2I_2NO_3$), ethyl bromoacetate (2 mmol), potassium carbonate (4 mmol), and tetrahydrofuran (20 mL) were added into a three-necked flask and then heated to reflux temperature for 24 h. The reaction was monitored via thin-layer chromatography (TLC). When it was complete, the mixture was concentrated to remove most of the solvent under reduced pressure, then poured into ice water, and the precipitate was collected by filtration and washed with water; compound **b** was obtained as a light yellow solid. Without further purification, the product was used in the next reaction.

The second step was to hydrolyze compound **b** into compound **c** ($C_{21}H_{13}Cl_2I_2NO_5$). Compound **b** (0.5 mmol) from the above step was mixed with ethanol and sodium hydroxide aqueous solution (20 mL). The mixture was stirred at room temperature until the disappearance of the starting material was detected by TLC. The reaction solution was concentrated *in vacuo*, followed by extraction with n-hexane to remove compound **b**. The aqueous layer was adjusted to acidic with hydrochloric acid. Finally, the precipitate was filtered and dried under vacuum. Compound **c,** dikalisuan, was obtained as a white solid. The synthesis route is shown in Fig. 1.

## Analytical methods

Mass spectra of the target compounds were recorded on a Dionex Ultimate 3000 with a Q-Exactive-Orbitrap mass spectrometer (Thermo Fisher Scientific, Bremen, Germany). [1]H NMR and [13]C NMR spectra were obtained using a Bruker DPX-400 spectrometer in the DMSO-$d_6$ solution. The purity of the dikalisuan was assessed by high-performance liquid chromatography (HPLC) on an Agilent HPLC system with a variable wavelength detector (VWD) (282 nm). The chromatographic separation was performed with a reverse-phase C18 analytical column (150 × 4.6 mm inner diameter, 5 µm particle size; Acchrom Technologies Co., Ltd.; Beijing, China) maintained at 30℃. The mobile phase consisted of phosphoric acid solution (pH = 3.0) and acetonitrile (v:v = 8:92), whose flow rate was set as 1.0 mL/min. The injection volume was 5 µL.

## Antimicrobial susceptibility test

Antimicrobial susceptibility assays using the microdilution broth method were per-formed according to the CLSI guidelines (18). Briefly, all drugs were two-fold serially diluted with Mueller-Hinton broth (MHB) in a sterile 96-well microtiter plate. Next, the log-phase bacteria suspensions were adjusted to $1 \times 10^6$ CFU/mL and mixed with drugs and then incubated at 37℃ for 18 h. Minimum inhibitory concentration (MIC) values were defined as the minimum concentration of the drug without visible growth of bacteria. Positive wells (with bacteria) or negative wells (without bacteria) were set as controls. Using the method above, add COL (2 µg/mL) to the bacterial suspension to determine the MIC of the dikalisuan in the tested strain, and define the MIC of the dikalisuan as the minimum COL-resistance reversal concentration (MRC) (19). All experiments were performed with three biological replicates.

## Checkerboard assay

The checkerboard assays were used to determine the synergistic activity of COL and dikalisuan in combination. Similar to the antimicrobial susceptibility test, COL was continuously diluted along the abscissa, while dikalisuan was continuously diluted along the ordinate in a 96-well microtiter plate. The bacterial suspension diluted with MHB was added into each well to achieve an ultimate bacterial concentration of approximately

**FIG 1** Synthesis strategy of RAF derivatization. (a) RAF, (b) esterification reaction of RAF, and (c) carboxymethylation reaction of RAF (dikalisuan).

$5 \times 10^5$ colony-forming unit (CFU)/mL. After incubating at 37 °C for 18 h, the fractional inhibitory concentration index (FICI) was calculated according to the following formula:

$$FICI = FICI_A + FICI_B = MIC_{AB}/MIC_A + MIC_{BA}/MIC_B.$$

$MIC_A$: the MIC of A alone, $MIC_B$: the MIC of B alone, $MIC_{AB}$: the MIC of A in combination with B, $MIC_{BA}$: the MIC of B in combination with A. FICI value ≤ 0.5 was defined as synergy. Three biological replicates were conducted for all experiments.

## Time-killing analysis

Time-killing experiments were conducted on *E. coli* HFE2171, *Salmonella* SH134, *K. pneumoniae* 3F, and *A. baumannii* 18D to further characterize the synergistic activity of COL and dikalisuan. We conducted the time-killing curve assays by combining three selected concentrations of dikalisuan (2, 8, and 32 µg/mL) with COL at 1/2 MIC. The monotherapy group was administered 1/2 MIC of COL or 32 µg/mL of dikalisuan. Samples and a control group (drug-free) were simultaneously incubated with shaking at 37°C. Bacterial cells with a final concentration of $1 \times 10^5$ CFUs/mL were cultured in MHB, COL, dikalisuan, and COL combined with dikalisuan. One hundred microliters of bacterial cells was collected from each group at 0, 2, 4, 8, 12, and 24 h, diluted with PBS, and spotted on LB agar plates and incubated overnight at 37°C to count bacterial colonies. Each assay was performed in triplicates. The synergy of combination therapy is defined as a reduction in bacteria of ≥2 $\log_{10}$ CFU/mL compared with the most active drug alone.

## Safety assessment

The hemolytic activity of dikalisuan and its combination with COL were determined according to the previous report (20). We collected red blood cells of mice by centrifugation. Then, mice blood cells were treated with different concentrations of dikalisuan (2–128 µg/mL) and its combination with COL (2 µg/mL) at 37°C for 1 h. Pure water was used as a positive control. Measure the absorbance of the supernatant at 540 nm and calculate the hemolysis of each sample by comparing it with a positive control.

The CCK-8 method was adopted to detect the cytotoxicity of dikalisuan (2–128 µg/mL) and its combination with COL (2 µg/mL). Generally, 100 µL of PK-15 cells (porcine kidney epithelial cells) was distributed into a 96-well plate. After overnight incubation, different concentrations of dikalisuan (2–128 µg/mL) or in combination with 2 µg/mL COL were added, and normal cultured cells were used as the control. After 24-h incubation, the cells were treated with 10 µL of CCK-8 and incubated for another 2 h. Finally, the viability of PK-15 cells was measured by recording and calculating based on the absorbance at 450 nm using a microplate reader (21). All experiments were performed in triplicate.

## Statistical analysis

Statistical analysis was performed using SPSS version 26.0 and GraphPad Prism version 8.0 software. All data from at least three biological replicates are presented as the mean ± SD. Unpaired *t*-tests were used to calculate *P*-values. Statistical significance

was determined at *P < 0.05, **P < 0.01, and ***P < 0.001; ns denotes no significant difference.

## RESULTS

### NMR and HRMS analysis of ethyl 2-(2-((3-chloro-4-(4-chlorophenoxy)phenyl)carbamoyl)-4,6-diiodophenoxy)acetate

Light yellow solid; Yield: 81.3%; $^1$H NMR(400 MHz, DMSO $d_6$): δ 10.71 (s, 1H), 8.31 (d, $J$ = 2.1 Hz, 1H), 8.02 (d, $J$ = 2.5 Hz, 1H), 7.88 (d, $J$ = 2.1 Hz, 1H), 7.59 (dd, $J$ = 8.9, 2.5 Hz, 1H), 7.45–7.38 (m, 2H), 7.24 (d, $J$ = 8.9 Hz, 1H), 6.98–6.91 (m, 2H), 4.64 (s, 2H), 4.09 (q, $J$ = 7.1 Hz, 2H), and 1.14 (t, $J$ = 7.1 Hz, 3H). $^{13}$C NMR(101 MHz, DMSO $d_6$): δ 167.87, 165.26, 156.47, 154.94, 148.51, 147.00, 137.89, 136.87, 133.43, 130.32, 127.24, 125.53, 123.02, 121.98, 120.75, 118.72, 95.90, 90.82, 62.53, 61.24, and 14.38. HRMS calculated for $C_{23}H_{17}Cl_2I_2NO_5[M + H]^+$: 711.8646, found: 711.8593. The HRMS and NMR spectra of compound **b** ($C_{23}H_{17}Cl_2I_2NO_5$) are shown in Fig.S1, S3, and S4.

### 2-(2-((3-chloro-4-(4-chlorophenoxy)phenyl)carbamoyl)-4,6-diiodophenoxy)acetic acid

White solid; Yield: 85.6%; $^1$H NMR(400 MHz, DMSO $d_6$): δ 13.05 (s, 1H), 10.69 (s, 1H), 8.31 (d, $J$ = 2.1 Hz, 1H), 8.01 (d, $J$ = 2.5 Hz, 1H), 7.90 (d, $J$ = 2.1 Hz, 1H), 7.61 (dd, $J$ = 8.9, 2.5 Hz, 1H), 7.46–7.39 (m, 2H), 7.23 (d, $J$ = 8.9 Hz, 1H), 6.99–6.91 (m, 2H), and 4.56 (s, 2H). $^{13}$C NMR(101 MHz, DMSO $d_6$): δ 169.27, 163.37, 156.45, 154.98, 148.59, 146.96, 137.99, 136.88, 133.38, 130.32, 127.20, 125.49, 122.95, 122.13, 120.89, 118.72, 96.05, 90.45, and 70.85. HRMS calculated for $C_{21}H_{13}Cl_2I_2NO_5[M + H]^+$: 683.8333, found: 683.8328. HPLC purity: 95.8%, HPLC $t_R$: 14.3 min. The HRMS and NMR spectra of dikalisuan ($C_{21}H_{13}Cl_2I_2NO_5$) are shown in Fig. S2, S5, and S6. HPLC is used to detect dikalisuan, as shown in Fig. S7.

### Synergistic antibacterial effects of the RAF derivative on COL-resistant gram-negative bacteria

The results of the antimicrobial susceptibility test and synergy after carboxymethylation of RAF are shown in Table 1. We determined the antibacterial effect of dikalisuan and COL alone, as well as their combination, against COL-resistant gram-negative bacteria. As shown in the detailed data in Table 1, among the COL-resistant strains, the MIC values of COL range from 4 to 64 µg/mL, while dikalisuan had no distinct inhibitory effect, with MIC values all exceeding 512 µg/mL. Interestingly, when both were used in

**TABLE 1** MIC, MRC, and FICI values of dikalisuan and COL

| Strains | | mcr-1 | Monotherapy MIC (µg/mL) | | Combination MIC (µg/mL) | | MRC (µg/mL) | FICI |
|---|---|---|---|---|---|---|---|---|
| | | | COL | Dikalisuan | COL | Dikalisuan | Dikalisuan | |
| E. coli | HFE2171 | + | 4 | >512 | 0.5 | 4 | 2 | 0.1328 |
| | AFE2146 | | 8 | >512 | 0.5 | 2 | 2 | 0.0664 |
| | AFE2131 | – | 4 | >512 | 0.5 | 2 | 2 | 0.1289 |
| | JCE2119 | | 4 | >512 | 0.5 | 2 | 2 | 0.1289 |
| Salmonella | SH12 | + | 4 | >512 | 0.5 | 4 | 2 | 0.1328 |
| | SH134 | | 4 | >512 | 0.5 | 8 | 1 | 0.1406 |
| | S2a | – | 4 | >512 | 0.125 | 8 | 0.5 | 0.0469 |
| | 11R | | 4 | >512 | 0.125 | 16 | 1 | 0.0625 |
| K. pneumoniae | 3F | + | 8 | >512 | 0.5 | 4 | 2 | 0.0703 |
| | 3Q | | 4 | >512 | 0.25 | 8 | 0.25 | 0.0781 |
| | 7G | | 64 | >512 | 0.5 | 4 | 1 | 0.0156 |
| | KP1 | – | 8 | >512 | 0.125 | 8 | 2 | 0.0313 |
| | KP9 | | 4 | >512 | 0.25 | 4 | 2 | 0.0703 |
| | E4 | | 16 | >512 | 0.25 | 4 | 4 | 0.0234 |
| A. baumannii | 18D | – | 8 | >512 | 0.25 | 8 | 2 | 0.0469 |

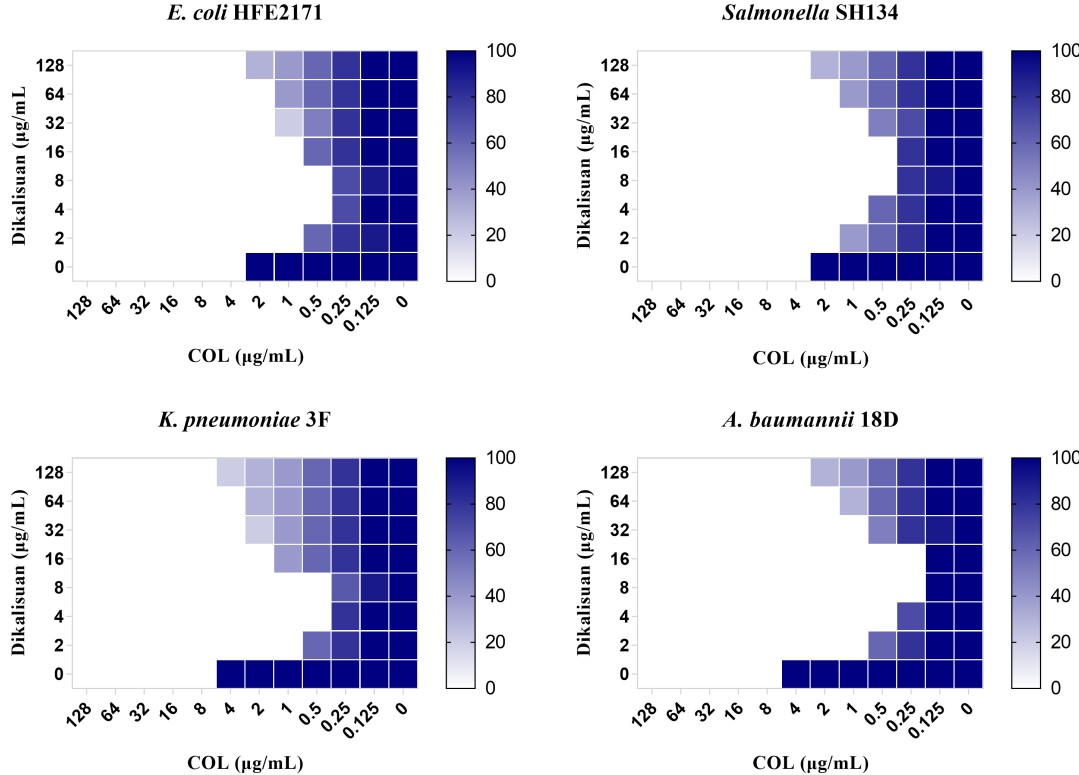

**FIG 2** Checkerboard assays between dikalisuan and COL against *E. coli* HFE2171, *Salmonella* SH134, *K. pneumoniae* 3F, and *A. baumannii* 18D.

combination, excellent synergy was exhibited, with FICI values ranging from 0.0234 to 0.1406 (all less than 0.5, as shown in Table 1 and Fig. 2). According to FICI values, the combination of dikalisuan and COL exhibits good synergistic antibacterial activity on COL-resistant gram-negative bacteria. In addition, when dikalisuan was combined with COL of 2 µg/mL, the MRC values of dikalisuan were 0.25–4 µg/mL (Table 1). Compared with the single drug, 2 µg/mL of COL reduced the MIC of dikalisuan by at least 128 times, which further indicated the synergy against gram-negative bacteria. The significance of MRC is to obtain the concentration of dikalisuan required for the transformation of MIC from resistance to sensitivity, in order to further demonstrate the synergistic effect.

To confirm the synergistic effect mentioned above, we selected four different types of gram-negative bacteria and used different concentrations of dikalisuan in combination with COL to perform the time-kill curve assays. As shown in Fig. 3, neither the sole use of dikalisuan nor of COL (1/2 MIC) alone could inhibit the growth of the tested strains within 24 h (ns, no significant difference). Overall, the two exhibited synergistic bactericidal efficacy, while dikalisuan showed a nonconcentration-dependent effect in the combination therapy group, with moderate concentrations (8 µg/mL) showing better synergistic antibacterial activity ($P < 0.05$) than low and high concentrations (2 µg/mL and 32 µg/mL, respectively). At moderate concentrations, it effectively killed *E. coli* HFE2171, *Salmonella* SH134, *K. pneumoniae* 3F, and *A. baumannii* 18D within 8 h. However, it only exhibited bacteriostasis within a certain period at high and low concentrations. The reasons for this phenomenon require further investigation into its underlying mechanism.

### *In vitro* safety of the drug alone and in combination with COL

Ensuring safety and lack of toxicity are essential themes in drug development. To further evaluate the potential of dikalisuan as an antibacterial adjuvant for COL, we assessed the safety of dikalisuan and its combination with COL on red blood cells of mice and mammalian kidney cells of PK-15 (Fig. 4). We found that dikalisuan did not cause hemolysis of red blood cells within the concentration range of 2–128 µg/mL (their

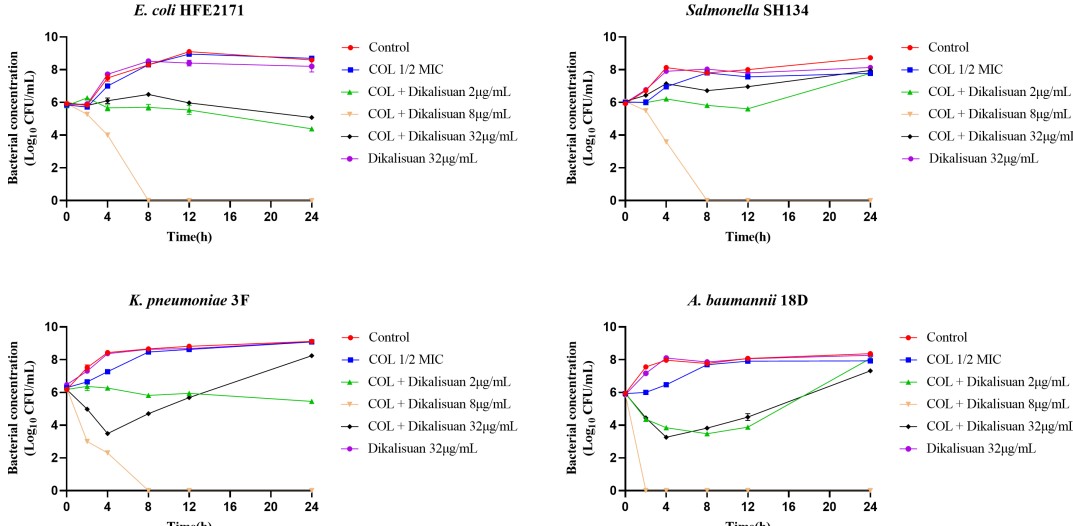

**FIG 3** Antibacterial activity of dikalisuan in combination with COL against *E. coli* HFE2171, *Salmonella* SH134, *K. pneumoniae* 3F, and *A. baumannii* 18D.

hemolysis rates were all below 5%). Moreover, at this concentration range, dikalisuan showed no significant cytotoxicity to PK-15 cells. The experimental $CC_{50}$ (compound's concentration required for the reduction of cell viability by 50%) for the PK-15 cell line was higher than 128 µg/mL. Compared with the single drug, when COL (2 µg/mL) was used in combination, the hemolysis rate increased, but no red blood cell hemolysis occurred (below 5%). Similarly, the combined use of dikalisuan did not show significant cytotoxicity even at 128 µg/mL. We preliminarily speculate that the combined group experienced a decline in safety due to the presence of COL, but there is no significant cytotoxicity, indicating that dikalisuan has the potential to be used as an antibacterial adjuvant for COL.

## DISCUSSION

We are currently in the "post-antibiotic era," with few clinical antibacterial candidates, especially for gram-negative pathogens (22). Although recognized as a last-resort antibiotic for treating infections caused by gram-negative bacteria, the therapeutic efficacy of COL is significantly diminished by MCR-mediated COL resistance (23). However, in the most recent years, the extensive horizontal spread of the MCR family has compelled us to seek an alternative approach as a treatment strategy to combat COL resistance (8). Given the long development cycle and high failure rate of antibacterial drugs, the combined treatment with existing drugs is the key to combating multidrug-resistant infections and extending their shelf lives (24). In the current study, the common types of adjuvants used with COL include natural compounds, antibiotics,

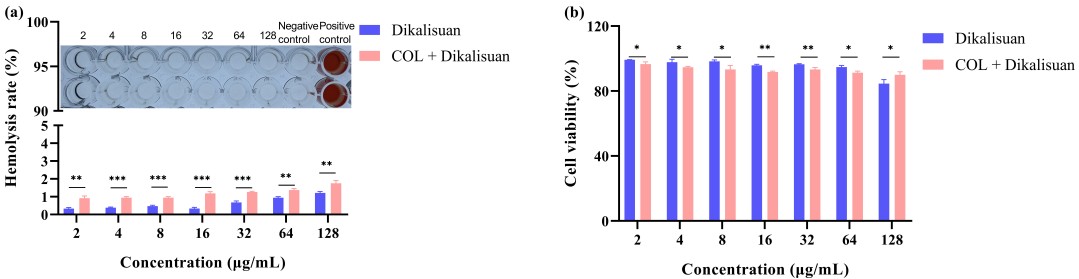

**FIG 4** Safety evaluation of dikalisuan and its combination with COL. (a): Hemolytic activity of different concentrations of dikalisuan or in combination with 2 µg/mL COL to mice red blood cells. (b): Cytotoxicity of the PK-15 cell line treated with different concentrations of dikalisuan or in combination with 2 µg/mL COL was tested based on the CCK-8 assay. *$P < 0.05$, **$P < 0.01$, and ***$P < 0.001$.

and non-antibiotic drugs (25–27). Li et al. reported that the derivative of polymyxin exhibits excellent antibacterial activity and also has relatively low toxicity to the HK-2 cell line (28). They carried out structural modifications to the antibiotic, while Luo et al. conducted related research on the adjuvant. A series of cationic poly(2-oxazoline)s with varying hydrophobic side chain lengths was synthesized via the controllable 2-oxazoline polymerization. Its combination with polymyxin B effectively enhanced the antibacterial activity (29). Yi et al. co-loaded COL and niclosamide (NIC) into mPEG-PLGA nanoparticles (COL/NIC-mPEG-PLGA-NPs) to overcome the resistance of multiple COL-resistant bacteria to COL (30). By simultaneously encapsulating antibiotic and adjuvant, it is possible to deliver these two drugs with vastly different pharmacokinetic properties to the site of infection in sufficient amounts.

In this study, the chemical structure of the RAF was modified, and the carboxymethylation of RAF was carried out through a two-step process. The results of MRC and FICI indicate that the combination of the derivatized product with COL can effectively restore the sensitivity of COL-resistant gram-negative bacteria to COL, regardless of whether they are *mcr-1*-positive or -negative strains. The results of the time-killing experiments showed that at moderate concentrations (8 µg/mL), dikalisuan had bactericidal activity on four types of COL-resistant gram-negative bacteria. Moreover, both low and high concentrations of dikalisuan exhibited a certain degree of bacteriostasis on these strains. This further demonstrates the synergistic effect of COL and dikalisuan. Interestingly, we found that dikalisuan was not concentration-dependent in the combined treatment group, which is similar to the finding of Cui et al. on the reversal of COL resistance by closantol (CST) (19). High reversal efficiency can be achieved within a low CST concentration range. However, as the CST concentration increases, the ability to reverse COL resistance remains unchanged or decreases, resulting in a gradual decrease in reversal efficiency. These results indicate that the adjuvant enhancement effect requires precise control of the concentration range rather than blindly increasing the dose. This study provides a theoretical basis for the *in vivo* experiment of dikalisuan on COL resistance in humans or other animals. In the study by El-Banna et al., they compared the pharmacokinetic parameters of ivermectin alone and in combination with RAF. Ivermectin was rapidly absorbed by RAF induction, and its elimination half-life was significantly prolonged (31). In future work, more in-depth mechanism and PK-PD studies are needed to verify its dose-response relationship so as to provide a more scientific theoretical basis for the clinical application of dikalisuan and the development of appropriate drug delivery systems.

Although literature has reported that RAF can enhance the antibacterial activity of COL against *A. baumannii*, *P. aeruginosa*, and *K. pneumoniae*, the clinical application of anthelmintic salicylanilides is severely hindered by their poor water solubility, low bioavailability, lack of targeting, and other drawbacks (32, 33). The inherent good properties of nanotechnology can promote antibacterial activity against many microorganisms, including bacteria, viruses, and fungi, and also improve the solubility of insoluble drugs (34). Recently, advanced drug delivery systems have made considerable progress in the treatment of bacterial infections based on the nanoemulsion gel, liposomes, polymeric micelles, nanoemulsion, and nanoparticles (35–39). Zhang et al. prepared a self-nanoemulsifying drug delivery system (SNEDDS) coencapsulating NIC and COL, which effectively solved the problem of water solubility of NIC (40). However, COL has good water solubility and is generally located in the outer layer of the carrier after being encapsulated, resulting in its characteristic of being prone to leakage or release. Our next plan is to connect the carboxymethylated derivative of RAF with COL through amide bonding via an amidation reaction and then prepare them into nanomedicines to achieve optimal synergistic antibacterial activity.

In conclusion, dikalisuan was synthesized via two steps through the carboxymethylation of RAF, and its structure was verified by [1]H NMR, [13]CNMR, and HRMS. The findings of synergistic antibacterial activity of dikalisuan and COL against *Escherichia coli*, *Salmonella*, *Klebsiella pneumoniae*, and *Acinetobacter baumannii* suggest that dikalisuan has an

ideal synergistic effect on COL-resistant gram-negative bacteria. Not only does it provide more adjuvant options for clinical use, but it also serves as a reference for the chemical modification of anthelmintic salicylanilides.

## ACKNOWLEDGMENTS

We thank Professor Hu and Professor Yang for their detailed guidance on this research. This work was supported by research grants from the National Key Research and Development Program of China (2023YFD1800105) and the National Natural Science Foundation of China (32373069).

ASM does not own the copyrights to Supplemental Material that may be linked to, or accessed through, an article. The authors have granted ASM a non-exclusive, world-wide license to publish the Supplemental Material files. Please contact the corresponding author directly for reuse.

## AUTHOR AFFILIATIONS

[1]College of Veterinary Medicine, Henan Agricultural University, Zhengzhou, Henan, China
[2]Shangqiu Meilan Biological Engineering Co., LTD, Shangqiu, Henan, China
[3]College of Sciences, Henan Agricultural University, Zhengzhou, Henan, China

## AUTHOR ORCIDs

Yajun Zhai http://orcid.org/0009-0002-2218-4819
Hua Wu https://orcid.org/0000-0002-1301-3076
Dandan He https://orcid.org/0000-0002-1536-6465
Guoyu Yang http://orcid.org/0009-0003-5377-3955
Gongzheng Hu http://orcid.org/0000-0002-7042-5771

## FUNDING

| Funder | Grant(s) | Author(s) |
| --- | --- | --- |
| National Key Research and Development Program of China | 2023YFD1800105 | Gongzheng Hu |
| National Natural Science Foundation of China | 32373069 | Gongzheng Hu |

## ADDITIONAL FILES

The following material is available online.

### Supplemental Material

**Supplemental figures (Spectrum03612-25-S0001.pdf).** Fig. S1 to S7.

### Open Peer Review

**PEER REVIEW HISTORY (review-history.pdf).** An accounting of the reviewer comments and feedback.

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
