## [Reviewer comments · Microbiology Spectrum]

Microbiology Spectrum

Synthesis and synergistic antibacterial activity of rafoxanide derivative as a novel drug for potentiating colistin activity

Haotian Shao, Qiange Liu, Peiyi Liu, Xiaoyuan Ma, Zibo Li, Ya-jun Zhai, Hua Wu, Dandan He, Guoyu Yang, and Gongzheng Hu

Corresponding Author(s): Gongzheng Hu, Henan Agricultural University

Review Timeline:

Submission Date:	November 9, 2025
Editorial Decision:	December 29, 2025
Revision Received:	January 30, 2026
Accepted:	February 22, 2026

Editor: Haifang Zhang

Reviewer(s): Disclosure of reviewer identity is with reference to reviewer comments included in decision letter(s). The following individuals involved in review of your submission have agreed to reveal their identity: Yujie Hu (Reviewer #1); Innocent Afeke (Reviewer #2)

Transaction Report:

DOI: <https://doi.org/10.1128/spectrum.03612-25>

Re: Spectrum03612-25 (Synthesis and synergistic antibacterial activity of rafoxanide derivative as a novel drug for reversing colistin resistance)

Dear Mr. GongZheng Hu:

Thank you for the privilege of reviewing your work. Below you will find my comments, instructions from the Spectrum editorial office, and the reviewer comments.

Revision Guidelines

Sincerely,
Haifang Zhang
Editor
Microbiology Spectrum

Reviewer #1 (Comments for the Author):

The authors synthesized a derivative, dikalisuan, from the anthelmintic rafoxanide (RAF) and confirmed its structure via NMR and HRMS. They investigated its synergistic antibacterial activity with colistin (COL) against *Escherichia coli*, *Salmonella*, *Klebsiella pneumoniae*, and *Acinetobacter baumannii*. The results indicate that dikalisuan exhibits ideal synergistic effects. This is indeed an excellent case study. As the authors state, the research provides additional options for clinical applications and

offers references for chemical modifications of anthelmintic benzamide compounds. Overall, I find this manuscript exceptionally well-written, concise yet comprehensive, with no superfluous content. The introduction, in particular, represents an ideal model in my view. The authors clearly present the rationale, methodology, and outcomes, supported by thorough discussion and robust data. This is one of the few manuscripts I have encountered with minimal concerns. I only have minor suggestions:

(1) The CLSI version cited (2021) is outdated. Given annual updates, the authors should update this reference and verify if other sections require corresponding revisions.

(2) The inclusion of both *mcr-1*-positive and -negative strains is commendable. However, note that 10 *mcr* gene types exist, each with multiple variants. Additionally, only strains with COL MICs of 4 µg/mL and 8 µg/mL (near the resistance breakpoint) were tested. Strains with higher MICs (e.g., 64/128/256/512 µg/mL, indicating stronger resistance) and *mcr-1*-negative strains with alternative COL resistance mechanisms should be included to broaden the validation of combination efficacy.

(3) The finding that dikalisuan's synergy in combination therapy is not concentration-dependent-requiring precise concentration control rather than dose escalation-is highly significant. While this does not undermine the core discovery, further R&D and clinical evaluation are needed. The authors should expand the discussion with examples of similar cases and elaborate on their perspectives, hypotheses, and future plans.

(4) Microbiology Spectrum requires Research Articles to be ~5,000 words (excluding Materials and Methods, References, tables, and figure legends). The current manuscript falls short. The authors and editor should decide whether to increase word count or consider an alternative article type. If expanding content, suggestions include: addressing the above points, adding details on *mcr* variants and COL resistance mechanisms, and incorporating comparisons with related studies in the Introduction and Discussion.

Reviewer #2 (Comments for the Author):

Major Comments

1. Terminology and Conceptual Accuracy

The manuscript repeatedly uses the phrase "reversing colistin resistance"; however, the experimental endpoints presented (MIC reduction and FICI-based synergy) do not demonstrate restoration of susceptibility according to CLSI or EUCAST clinical breakpoints. Based on the reported MRC values (0.25-2 µg/mL), the findings indicate potentiation of colistin activity rather than true resistance reversal. I therefore suggest that the authors revise the title, abstract, results, and discussion to reflect terms such as "potentiates colistin activity" or "enhances colistin efficacy", rather than implying clinical resistance reversal, which is not supported by the data.

2. Incomplete Bacterial Strain Characterization

The Methods section refers to *mcr-1*-positive and *mcr-1*-negative isolates; however, essential details required for reproducibility and interpretation are missing. Could the authors specify the number of isolates tested per species? In addition, for *mcr-1*-negative isolates, the underlying mechanisms of colistin resistance (e.g., *pmrAB* or *phoPQ* mutations) are not described. Furthermore, clarification is needed on how *mcr-1* status was confirmed (PCR, sequencing, or prior characterization). I suggest that these details be included, as they are critical for contextualizing the observed synergistic effects.

3. Methodological Clarity: MRC and Time-Kill Assays

The use of the MRC endpoint, which is not recognized by CLSI or EUCAST standards, requires clearer definition and justification. Could the authors explicitly describe how MRC was determined and discuss its relevance in the context of established antimicrobial susceptibility testing frameworks?

Similarly, the description of time-kill assays lacks important methodological details. Specifically, the manuscript does not clearly report the exact drug concentrations used in each experiment, the number of biological or technical replicates performed, or the statistical treatment of bacterial count data. Without these details, quantitative interpretation of synergistic and bactericidal effects is limited.

4. Statistical Analysis

No statistical methods are described in the manuscript. Could the authors clarify which statistical tests were used, how variability was handled, and what criteria were applied to determine significance? Even for *in vitro* assays, particularly time-kill studies, basic statistical analysis is necessary to support claims such as concentration-dependent effects and comparative activity.

5. Overinterpretation of Results

The Results and Discussion sections frequently use terms such as "therapeutic effect" and present species-specific rankings (e.g., "better effect on *Klebsiella pneumoniae* and *Acinetobacter baumannii*"), which are not supported by *in vitro* data alone. I suggest that the authors limit their conclusions to antibacterial or synergistic activity observed under laboratory conditions. In addition, speculation regarding mechanisms underlying concentration-dependent effects and proposed nanomedicine applications (lines 236-259) extends beyond the scope of the presented experiments. For example, while higher dikalisuan concentrations showed limited bacteriostasis, mechanistic interpretations should be framed as hypotheses and deferred to future investigations.

6. Compound Purity Assessment

Although NMR and HRMS data confirm compound identity, no compound purity assessment (e.g., HPLC analysis) is provided. Could the authors include purity data, given that the compound was used for biological assays? This information is important for reproducibility and accurate interpretation of antimicrobial potency.

Minor Comments

•Replace the term "therapeutic effect" with "bactericidal effect" or "synergistic antibacterial activity" throughout the manuscript.

- Ensure that FICI values and MRC data are reported as mean {plus minus} standard deviation, with the number of independent replicates clearly stated.
- The literature review on nanomedicine-based delivery systems could be condensed to maintain focus on the current experimental findings.

Dear Prof. Haifang Zhang and reviewers,

We are truly grateful for your positive and constructive comments and suggestions on our manuscript entitled “Synthesis and synergistic antibacterial activity of rafoxanide derivative as a novel drug for reversing colistin resistance”. Based on these comments and suggestions, we have carefully modified the original manuscript. All changes made to the manuscript are highlighted in yellow.

Our point-to-point responses to the comments from the reviewers are listed below. We hope this revision meets MS’s standards and receives some positive comments.

Yours sincerely,

Gongzheng Hu, PhD

Detailed Responses to Referee 1

The authors synthesized a derivative, dikalisuan, from the anthelmintic rafoxanide (RAF) and confirmed its structure via NMR and HRMS. They investigated its synergistic antibacterial activity with colistin (COL) against *Escherichia coli*, *Salmonella*, *Klebsiella pneumoniae*, and *Acinetobacter baumannii*. The results indicate that dikalisuan exhibits ideal synergistic effects. This is indeed an excellent case study. As the authors state, the research provides additional options for clinical applications and offers references for chemical modifications of anthelmintic benzamide compounds. Overall, I find this manuscript exceptionally well-written, concise yet comprehensive, with no superfluous content. The introduction, in particular, represents an ideal model in my view. The authors clearly present the rationale, methodology, and outcomes, supported by thorough discussion and robust data. This is one of the few manuscripts I have encountered with minimal concerns. I only have minor suggestions.

Response: Thanks very much for your positive comments.

(1) The CLSI version cited (2021) is outdated. Given annual updates, the authors should update this reference and verify if other sections require corresponding revisions.

Response: Thank you very much for your suggestion. We have updated the reference and verified other sections in the manuscript (Lines 425-427).

(2) The inclusion of both *mcr-1*-positive and -negative strains is commendable. However, note that 10 *mcr* gene types exist, each with multiple variants. Additionally, only strains with COL MICs of 4 µg/mL and 8 µg/mL (near the resistance breakpoint) were tested. Strains with higher MICs (e.g., 64/128/256/512 µg/mL, indicating stronger resistance) and *mcr-1*-negative strains with alternative COL resistance mechanisms should be included to broaden the validation of combination efficacy.

Response: Thank you very much for your valuable advice. Because the *mcr-1* is the first plasmid-mediated COL resistance gene discovered and widely spread, we are focusing on it. We added two strains of *K. pneumoniae*, one *mcr-1* positive (7G, mic = 64 µg/mL) and the other negative (E4, mic = 16 µg/mL), and included the results in **TABLE 1**.

(3) The finding that dikalisuan's synergy in combination therapy is not concentration-dependent-requiring precise concentration control rather than dose escalation-is highly significant. While this does not undermine the core discovery, further R&D and clinical evaluation are needed. The authors should expand the discussion with examples of similar cases and elaborate on their perspectives, hypotheses, and future plans.

Response: Thank you for your suggestion. Our research results are similar to the findings of Cui et al. on the reversal of COL resistance by closantol (CST). High reversal efficiency can be achieved within a low CST concentration range. However, as the CST concentration increases, the ability to reverse COL resistance remains unchanged or decreases, resulting in a gradual decrease in reversal efficiency. These results indicate that the adjuvant enhancement effect requires precise control of the concentration range rather than blindly increasing the dose. This study provides a theoretical basis for the in vivo experiment of dikalisuan reversing COL resistance in humans or other animals. In future work, more in-depth mechanism and PK-PD studies are needed to verify its dose-response relationship, so as to provide a more scientific theoretical basis for the clinical application of dikalisuan and the

development of appropriate drug delivery systems. We have added this content to the discussion section (Lines 306-315).

Cui XD, Liu SB, Wang RY, He DD, Pan YS, Yuan L, Zhai YJ, Hu GZ. 2024.

Investigation on the reversal effect of closantel on colistin resistance in MCR-1 positive *Escherichia coli* based on dose-response relationship. *J Antimicrob Chemother* 80:528-537. <https://doi.org/10.1093/jac/dkae441>.

(4) Microbiology Spectrum requires Research Articles to be ~5,000 words (excluding Materials and Methods, References, tables, and figure legends). The current manuscript falls short. The authors and editor should decide whether to increase word count or consider an alternative article type. If expanding content, suggestions include: addressing the above points, adding details on *mcr* variants and COL resistance mechanisms, and incorporating comparisons with related studies in the Introduction and Discussion.

Response: Thank you very much for your suggestion. Following your suggestion, we added detailed information about the *mcr* variants and the COL resistance mechanisms in the introduction section (Lines 65-80). Additionally, ensuring safety and lack of toxicity are essential themes in drug development, and we have added the haemolytic activity and cytotoxicity (PK-15) evaluation. It was described in the Materials and Methods section (Lines 182-198), and the results are shown in Figure 4 (Lines 526-530). The results showed that dikalisuan alone or in combination with COL did not cause red blood cell hemolysis or exhibit significant cytotoxicity (Lines 260-275), thereby providing a theoretical basis for the *in vivo* experiment of dikalisuan on COL resistance in humans or other animals.

Detailed Responses to Referee 2

Response: Thank you very much for your suggestions on this article.

Major Comments

1. Terminology and Conceptual Accuracy

The manuscript repeatedly uses the phrase "reversing colistin resistance"; however,

the experimental endpoints presented (MIC reduction and FICI-based synergy) do not demonstrate restoration of susceptibility according to CLSI or EUCAST clinical breakpoints. Based on the reported MRC values (0.25-2 µg/mL), the findings indicate potentiation of colistin activity rather than true resistance reversal. I therefore suggest that the authors revise the title, abstract, results, and discussion to reflect terms such as "potentiates colistin activity" or "enhances colistin efficacy", rather than implying clinical resistance reversal, which is not supported by the data.

Response: Thank you very much for your suggestion. Due to the incomplete presentation in **TABLE 1**, it may have caused your misunderstanding. We have added monotherapy and combination MIC groups to make the differences more obvious (Lines 515-518). According to the definition of COL in the CLSI guideline (2025), Sensitivity (S): MIC ≤ 2 µg/mL, Resistance (R): MIC ≥ 4 µg/mL. Our results showed that when COL was used in combination with dikalisuan, the MIC value of COL was 0.125-0.5 µg/mL, all less than 2 µg/mL. The transformation of MIC from resistance to sensitivity. The checkerboard assays and time-killing experiments indicated that the combination of the two drugs had a synergistic effect. In addition, MRC was defined by our laboratory according to the CLSI guideline (MIC ≤ 2 µg/mL, defining sensitivity), to further evaluate COL and adjuvants. Cui XD et al. evaluated closantel and COL using MRC. MRC was determined using standard broth microdilution method, and the specific operation steps are as follows: added COL (2 µg/mL) to the bacterial suspension to determine the dikalisuan MIC of the tested strain at this time, and defined this dikalisuan MIC as the minimum COL resistance reversal concentration (MRC). Using the description that enhances the activity of COL is more accurate. Therefore, we have revised the title and checked the other parts of the manuscript.

Cui XD, Liu SB, Wang RY, He DD, Pan YS, Yuan L, Zhai YJ, Hu GZ. 2024.

Investigation on the reversal effect of closantel on colistin resistance in MCR-1 positive *Escherichia coli* based on dose-response relationship. *J Antimicrob Chemother* 80:528-537. <https://doi.org/10.1093/jac/dkae441>.

2. Incomplete Bacterial Strain Characterization

The Methods section refers to *mcr-1*-positive and *mcr-1*-negative isolates; however, essential details required for reproducibility and interpretation are missing. Could the authors specify the number of isolates tested per species? In addition, for *mcr-1*-negative isolates, the underlying mechanisms of colistin resistance (e.g., *pmrAB* or *phoPQ* mutations) are not described. Furthermore, clarification is needed on how *mcr-1* status was confirmed (PCR, sequencing, or prior characterization). I suggest that these details be included, as they are critical for contextualizing the observed synergy effects.

Response: Thank you very much for your suggestion. All strains used in this study were collected and preserved in our laboratory. Our laboratory previously amplified the *mcr-1* gene through polymerase chain reaction (PCR) to define whether the *mcr-1* gene was detected as positive or negative. Because the *mcr-1* is the first plasmid-mediated COL resistance gene discovered and widely spread, we are focusing on it. We only amplified the *mcr* family genes of the KP1 and KP9 strains of *K. pneumoniae* using PCR, and the isolates were whole-genome sequenced. We found that they were chromosomally mediated resistance, and this was due to the inactivation of the *mgrB* gene. Other *mcr-1* negative strains may be mediated by genes from the remaining *mcr* family or by the chromosome. We have specified the number of isolates tested per species in the manuscript and clarified the determination of *mcr-1* status (Lines 101-108).

3. Methodological Clarity: MRC and Time-Kill Assays

The use of the MRC endpoint, which is not recognized by CLSI or EUCAST standards, requires clearer definition and justification. Could the authors explicitly describe how MRC was determined and discuss its relevance in the context of established antimicrobial susceptibility testing frameworks?

Similarly, the description of time-kill assays lacks important methodological details. Specifically, the manuscript does not clearly report the exact drug concentrations used in each experiment, the number of biological or technical replicates performed, or the statistical treatment of bacterial count data. Without these details, quantitative

interpretation of synergistic and bactericidal effects is limited.

Response: Thank you very much for your valuable advice. As stated in response to “Terminology and Conceptual Accuracy”: MRC was defined by our laboratory according to the CLSI guideline ($\text{MIC} \leq 2 \mu\text{g/mL}$ defining sensitivity), to further evaluate COL and dikalisuan have a synergistic antibacterial effect on COL-resistant Gram-negative bacteria. We have replaced the adjuvant in the MRC description with dikalisuan for a more accurate description. And further discussed MRC based on FICI values, explaining its significance for synergistic effect evaluation. (Lines 153; Lines 240-246). Similarly, we added the exact drug concentration, number of biological replicates performed and statistical treatment of bacterial count data (Lines 171-174; Lines 179; Lines 199-204).

4. Statistical Analysis

No statistical methods are described in the manuscript. Could the authors clarify which statistical tests were used, how variability was handled, and what criteria were applied to determine significance? Even for in vitro assays, particularly time-kill studies, basic statistical analysis is necessary to support claims such as concentration-dependent effects and comparative activity.

Response: Thank you for your suggestion. We have added the statistical analysis used in the experiment to the Materials and Methods section (Lines 199-204) and have also added the results of the time-kill studies (Lines 251; Lines 254).

5. Overinterpretation of Results

The Results and Discussion sections frequently use terms such as "therapeutic effect" and present species-specific rankings (e.g., "better effect on *Klebsiella pneumoniae* and *Acinetobacter baumannii*"), which are not supported by in vitro data alone. I suggest that the authors limit their conclusions to antibacterial or synergistic activity observed under laboratory conditions.

In addition, speculation regarding mechanisms underlying concentration-dependent effects and proposed nanomedicine applications (lines 236-259) extends beyond the scope of the presented experiments. For example, while higher dikalisuan concentrations showed limited bacteriostasis, mechanistic interpretations should be

framed as hypotheses and deferred to future investigations.

Response: Thank you for your suggestions. The results of the time-killing experiments showed that at moderate concentrations (8 µg/mL), dikalisuan had bactericidal activity on four types of COL-resistant Gram-negative bacteria. Moreover, both low and high concentrations of dikalisuan exhibited a certain degree of bacteriostasis on these strains. This further demonstrates the synergistic effect of COL and dikalisuan. We have revised the conclusion to focus on the aspect of synergistic effect (Lines 302-307).

For this part of the content, we have proposed hypotheses and envisioned the next steps of research (Lines 318-321).

6. Compound Purity Assessment

Although NMR and HRMS data confirm compound identity, no compound purity assessment (e.g., HPLC analysis) is provided. Could the authors include purity data, given that the compound was used for biological assays? This information is important for reproducibility and accurate interpretation of antimicrobial potency.

Response: Thank you very much for your suggestion. We detected the purity of dikalisuan (95.8%) by HPLC and added detailed detection conditions to the Analytical methods section (Lines 137-143), and purity information to the Results section (Lines 226). The HPLC chromatogram is shown in Figure S7.

Minor Comments

- Replace the term "therapeutic effect" with "bactericidal effect" or "synergistic antibacterial activity" throughout the manuscript.

Response: Done (Lines 254; Lines 337). Thanks.

- Ensure that FICI values and MRC data are reported as mean {plus minus} standard deviation, with the number of independent replicates clearly stated.

Response: Thank you for your suggestion. The MRC and FICI experiments in the manuscript were repeated at least three times each to ensure consistent results. Therefore, we did not use the average value to represent them. We have added information on the number of repetitions in the manuscript (Lines 154-155; Lines 167;

Lines 179).

•The literature review on nanomedicine-based delivery systems could be condensed to maintain focus on the current experimental findings.

Response: Thank you for your suggestion. We have moderately streamlined it. The main purpose of RAF derivatization is to couple it with COL through amidation reaction to reduce the adverse effects of COL. In addition, the prodrug formed by coupling COL with dikalisuan or the single drug of COL and dikalisuan can be prepared as nanomedicines to achieve passive targeting. This is our future work plan and outlook for dikalisuan. It is also possible that someone is engaged in the chemical modification of anthelmintic salicylanilides, and we hope that this research and work plan can provide some ideas for them. We hope this answer will meet your approval. Thank you very much.

Re: Spectrum03612-25R1 (Synthesis and synergistic antibacterial activity of rafoxanide derivative as a novel drug for potentiating colistin activity)

Dear Prof. Gongzheng Hu:

Your manuscript has been accepted, and I am forwarding it to the ASM production staff for publication. Your paper will first be checked to make sure all elements meet the technical requirements. ASM staff will contact you if anything needs to be revised before copyediting and production can begin. Otherwise, you will be notified when your proofs are ready to be viewed.

Sincerely,
Haifang Zhang
Editor
Microbiology Spectrum

Reviewer #1 (Comments for the Author):

The authors have perfectly addressed all the issues I previously raised, I have no further comments and recommend acceptance of this manuscript. BTW, from their carefully prepared response file, I also had the opportunity to review the other reviewer's comments and the authors' replies. I found this exchange highly insightful. I would like to express my appreciation to the other reviewer for their precise and valuable input, and my gratitude to the authors for diligently addressing and responding to all points.

Reviewer #2 (Comments for the Author):

The authors have substantially revised the manuscript and addressed many methodological and conceptual concerns I had. However, a clearer framing of susceptibility restoration as an in vitro observation is recommended.